# SEAL-Pose: Enhancing Pose Estimation through Trainable Loss Function

## Abstract

Accurately predicting 3D human pose is a challenging task in computer vision due to the need to capture complex spatial structures and anatomical constraints. We propose SEAL-Pose, an adaptation of the Structured Energy As Loss (SEAL) framework for deterministic models, specifically designed to enhance 3D human pose estimation from 2D keypoints. Although the original SEAL was limited to probabilistic models, our approach employs the model's predictions as negative examples to train a structured energy network, which functions as a dynamic and trainable loss function. Our approach enables a pose estimation model to learn joint dependencies via learning signals from a structured energy network that automatically captures body structure during training without explicit prior structural knowledge, resulting in more accurate and plausible 3D poses . We introduce new evaluation metrics to assess the structural consistency of predicted poses, demonstrating that SEAL-Pose produces more realistic, anatomically plausible results. Experimental results on the Human3.6M and Human3.6M WholeBody datasets show that SEAL-Pose not only reduces pose estimation errors such as Mean Per Joint Position Error (MPJPE) but also outperforms existing baselines. This work highlights the potential of applying structured energy networks to tasks requiring complex output structures, offering a promising direction for future research.

## 1 Introduction

Pose estimation is a critical task in computer vision that requires accurately predicting the keypoint positions of objects, such as humans. In particular, 3D human pose estimation (3D HPE) is even more challenging because it involves predicting spatial structures while adhering to anatomical constraints. (Liu et al., 2024) Despite recent advances, there remain significant limitations in methods' ability to effectively capture dependencies in the output space, which is critical for producing accurate and plausible 3D poses.

To address this issue, we propose an extension of the Structured Energy As Loss (SEAL) framework (Lee et al., 2022) to improve 3D HPE. SEAL leverages a structured energy network as a trainable loss function, allowing the model to learn dependencies among output variables. Although SEAL was originally designed for structured prediction tasks involving probabilistic models, we adapt the framework for deterministic models. This adaptation not only enhances 3D HPE but also broadens SEAL's applicability to various tasks that require learning complex output dependencies.

Our primary contribution is the adaptation of the SEAL for deterministic models for 3D HPE, particularly in a 2D-to-3D lifting scenario. The original SEAL framework utilized the output distribution of a neural network, referred to as a task-net, as a dynamic noise distribution to train a structured energy network, referred to as a loss-net, and thus could only be applied to probabilistic models. However, we successfully applied SEAL to deterministic models that directly output real-valued predictions by properly utilizing the task-net's output values as negative examples

Our proposed method SEAL-Pose, integrating the SEAL framework to 3D HPE, enables pose estimation models to learn and adapt the relationships between joints during training. SEAL-Pose allows the model to more accurately represent dependencies in the output space, improving 3D HPE's accuracy and lowering essential error metrics like Mean Per Joint Position Error (MPJPE). In addition, SEAL-pose leads to more plausible poses even without explicit prior knowledge about the structure of a given dataset or the human pose. There have been previous studies on 3D HPE

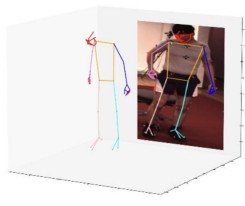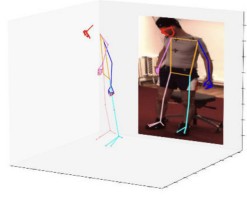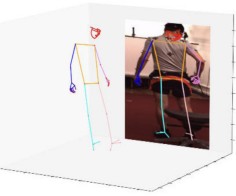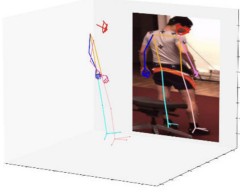

Figure 1: The example annotations of H3WB dataset

that reflect the structure of human pose well through manual encoding of body structure or domain-specific rules (Zheng et al., 2020; Wu et al., 2022; Fang et al., 2018; Xu et al., 2022). In contrast to existing methods, SEAL-Pose provides a dynamic loss function that reflects on joint dependencies automatically, as it is being trained. This provides a flexible and scalable method for 3D HPE by enabling more precise and consistent 3D pose predictions without the need for predetermined structural priors.

Additionally, we introduce new metrics to evaluate the structural consistency of predicted poses. These metrics highlight the advantage of our SEAL-based approach in producing more realistic, anatomically plausible results, even without explicit prior knowledge or constraints. Our experiments show that this adaptation of SEAL not only improves performance but also enables the model to learn more coherent output structures. This suggests that the SEAL-based approach could be applied effectively to a wide range of tasks where capturing complex dependencies and producing structured outputs are important.

## 2    RELATED WORK

### 2.1    STRUCTURED ENERGY AS LOSS (SEAL)

SEAL builds on the concept of using structured energy networks for structured prediction, initially introduced by (Belanger & McCallum, 2016) and extended by (Gygli et al., 2017). These early models, known as Structured Prediction Energy Networks (SPENs), effectively captured dependencies among output variables but were limited by slow and unstable inference due to reliance on gradient-based inference (GBI).

SEAL addresses these issues by using structured energy networks as trainable loss functions rather than direct predictors, enabling faster and more stable inference at test time. This approach has been applied to relatively straightforward tasks such as multi-label classification, semantic role labeling, and image segmentation, highlighting SEAL's potential to improve performance and efficiency over traditional methods. However, SEAL is limited in its applicability as it has only been applied to probabilistic models and has not been used with deterministic models.

### 2.2    3D HUMAN POSE ESTIMATION

3D human pose estimation is a well-established computer vision task involving the prediction of 3D joint locations from 2D images or videos. This task is inherently challenging, as it requires inferring spatial relationships and ensuring anatomical plausibility using limited visual information. Current approaches typically follow two paradigms: (1) directly predicting 3D poses from images or (2) using a two-step process where 2D poses are estimated first and then "lifted" to 3D space (Zheng et al., 2023; Liu et al., 2024). The latter approach, due to its reliance on the accuracy of 2D pose estimation, has become more popular and effective, driven by advances in 2D keypoint detection (Zheng et al., 2023). Therefore, we also adopted the 2D-to-3D lifting approach in our work.

3D whole-body pose estimation extends traditional 3D human pose estimation by integrating detailed annotations for additional keypoints, including those for the face, hands, and feet, enabling more fine-grained and precise applications. The expanded scope introduces greater challenges due to the variation in scales and the increased diversity of poses associated with these keypoints. Recently, (Zhu et al., 2023) developed the Human3.6M 3D WholeBody dataset (H3WB) based on

the widely used Human3.6M dataset (H36M) by including annotations for additional keypoints, as shown in Figure 1. This dataset has come out as an important resource, enabling research to address the increased complexity of whole-body pose estimation while encouraging methods that go beyond traditional approaches focused mainly on standard body keypoints.

Several works have focused on capturing the structural dependencies between body joints to improve pose estimation accuracy. For instance, (Zheng et al., 2020) proposed the Joint Relationship Aware Network, which enhances pose predictions by considering both global and local joint relationships. (Wu et al., 2022) introduced the Limb Poses Aware Network, which incorporates relative and absolute bone angles to model pose structure. However, these methods tend to be closely tied to specific model architectures. Another notable approach is Pose Grammar (Fang et al., 2018; Xu et al., 2022), which uses predefined kinematic rules and bidirectional recurrent neural networks to refine pose predictions. Despite its effectiveness, Pose Grammar relies on predefined knowledge of human body structure, which may hinder its scalability to new datasets or tasks where such information is unavailable or incomplete.

Our work builds on these foundations by introducing a novel application of the SEAL framework for 3D human pose estimation. Unlike existing methods that often require manual encoding of body structure or rely on domain-specific rules, SEAL-Pose offers a dynamic, trainable loss function that learns the dependencies between joints during training. This allows for more accurate and coherent 3D pose predictions without the need for predefined structural priors, offering a flexible and scalable approach to 3D pose estimation.

## 3 EXPERIMENTAL SETUP

### 3.1 SEAL-POSE

Our method, SEAL-Pose, adapts the SEAL framework for 3D human pose estimation, particularly in a 2D-to-3D lifting scenario. Unlike conventional methods that manually encode body structure or rely on domain-specific rules, SEAL-Pose uses a dynamic, trainable loss function to model joint dependencies. By incorporating the SEAL framework, our method allows any pose estimation architecture to better capture the relationships between joints, leading to more accurate and coherent 3D pose predictions.

In particular, we implement the SEAL-dynamic approach, in which the pose estimation model (task-net) and the structured energy network (loss-net) are trained jointly. In this framework, the task-net $F_\phi(x)$ is optimized to minimize a weighted sum of the Mean Squared Error (MSE) loss and the energy, output of the the loss-net, $E_\theta(x, \tilde{y})$. Specifically, the task-net parameters $\phi$ are updated using the following manner:

$$\phi_t \leftarrow \phi_{t-1} - \eta_\phi \nabla_\phi \frac{1}{|B_t|} \sum_{(x,y) \in B_t} L_F(\phi; \theta) \tag{1}$$

where $B_t$ is the mini-batch of training samples at iteration $t$, $\eta_\phi$ is the learning rate for the task-net, and $L_F(\phi; \theta)$ is the combined loss function. The combined loss function is defined as:

$$L_F(x_i, y_i; \theta) = \sum_{j=1}^{L} \text{MSE}(y_j, F_\phi(x)_j) + \alpha E_\theta(x, F_\phi(x)) \tag{2}$$

In this equation, $L$ refers to the total number of joints in the pose estimation task and $x$ represents the input data, specifically the 2D joint coordinates. The variable $y_j$ denotes the ground-truth 3D joint coordinates, while $F_\phi(x)_j = \tilde{y}_j$ represents the predicted 3D joint coordinates from the task-net. The energy term $E_\theta(x, F_\phi(x))$, computed by the loss-net, captures structural dependencies among joints. Finally, $\alpha$ is a hyperparameter controlling the balance between the MSE loss and the energy term. The loss-net is dynamically trained to adapt to the task-net's predictions by minimizing the energy loss $L_E$:

$$\theta_t \leftarrow \theta_{t-1} - \eta_\theta \nabla_\theta \frac{1}{|B_t|} \sum_{(x,y) \in B_t} L_E(x, y, F_{\phi_{t-1}}(x); \theta) \tag{3}$$

We select a margin-based loss for $L_E$, which enforces a margin between the true label $y$ and an incorrect prediction $\tilde{y}$:

$$L_E^{\text{margin}}(x_i, y_i, \tilde{y}_i; \theta) = \max_{\tilde{y}} \left[ \Delta(y, \tilde{y}) - E_\theta(x, \tilde{y}) + E_\theta(x, y) \right]_+ \tag{4}$$

Here, $\Delta(y, \tilde{y})$ denotes a task-specific margin function. This loss formulation encourages that the energy assigned to the correct label is lower than that of any incorrect prediction by a specified margin. In our implementation, we use the MPJPE as the margin function, and the weighting of energy terms is controlled via a hyperparameter.

Alternatively, we can employ a Noise-Contrastive Estimation (NCE) loss for $L_E$. The NCE loss trains the loss-net to assign lower energy to the true label compared to a negative sample $\tilde{y}$:

$$L_E^{\text{NCE}}(x_i, y_i, \tilde{y}_i; \theta) = -\log \frac{\exp(-E_\theta(x, y))}{\exp(-E_\theta(x, y)) + \exp(-E_\theta(x, \tilde{y}))} \tag{5}$$

Since our task-net does not output probability distributions, we cannot sample negative samples from a noise distribution. Instead, in both the margin-based loss and NCE loss cases, we use the task-net's predictions as $\tilde{y}$, allowing both the loss-net and task-net to be trained dynamically by leveraging the task-net's evolving predictions as negative samples. Additionally, in order to improve loss-net training without a noise distribution, we use a larger batch in updating the loss-net, which always includes the entire batch for the task-net.

In SEAL-dynamic, the task-net and loss-net are updated in alternative manner, allowing the loss-net to continuously adapt to the evolving state of the task-net, as shown in Algorithm 1. This iterative joint optimization process ensures that the loss-net remains synchronized with the task-net's progress, enhancing its ability to guide the task-net effectively. By dynamically modeling joint dependencies, this approach leads to more accurate and structurally consistent 3D pose predictions.

---

**Algorithm 1** SEAL-Pose Algorithm

---

**Require:** $(\mathbf{x}, \mathbf{y})$: training data (2D inputs and 3D ground-truth outputs)
**Require:** $F_\phi$: task-net with parameters $\phi$
**Require:** $E_\theta$: loss-net with parameters $\theta$
**Require:** optimizer$_\phi$, optimizer$_\theta$: optimizers for task-net and loss-net
**Require:** $T$: number of training iterations
1: Initialize $\phi_0, \theta_0$ randomly
2: **for** $t = 1$ to $T$ **do**
3:     Sample mini-batch $B_t = \{(x_i, y_i)\}_{i=1}^N$ from training data
4:     Compute task-net predictions: $\tilde{y}_i = F_{\phi_{t-1}}(x_i)$ for all $x_i \in B_t$
5:     Update loss-net parameters $\theta_t$:
6:         $\theta_t \leftarrow \theta_{t-1} - \eta_\theta \nabla_\theta \frac{1}{|B_t|} \sum_{(x_i, y_i) \in B_t} L_E(x_i, y_i, \tilde{y}_i \, \theta)$
7:     Update task-net parameters $\phi_t$:
8:         $\phi_t \leftarrow \phi_{t-1} - \eta_\phi \nabla_\phi \frac{1}{|B_t|} \sum_{(x_i, y_i) \in B_t} L_F(x_i, y_i; \theta_t)$
9: **end for**

---

## 3.2 GRADIENT-BASED INFERENCE

We implemented a gradient-based inference (GBI) (Lee et al., 2019) method as an additional baseline to evaluate the efficacy of utilizing a structured energy network as a direct predictor versus employing it as a loss network that provides a learning signal. GBI is a method that leverages gradients to iteratively refine the outputs or parameters of a neural network, progressively increasing

the likelihood that the output configuration will satisfy the desired constraints. In our case, the constraint is energy, the output of the energy network, must decrease. We specifically employed GBI to directly update the task-net's predictions using gradient signals derived from the energy network.

The implementation of GBI involves three main steps. The task-net, serving as our baseline model, is trained in a supervised manner to predict 3D poses. Next, a structured energy network is trained using the predictions from the task-net as negative samples. Lastly, the trained energy network is employed to iteratively update the task-net's predictions through gradient-based optimization.

---

**Algorithm 2** Gradient-Based Inference

---

**Require:** $(\mathbf{x}, \mathbf{y})$: training data (2D inputs and 3D ground-truth outputs)
**Require:** $F_\phi$: task-bet, $E_\theta$: energy network
**Require:** $\texttt{optimizer}_\theta$: optimizer for $E_\theta$
**Require:** $T$: training iterations, $K$: GBI steps
 1: **Phase 1: train Task-Net**
 2: **for** $t = 1$ to $T$ **do**
 3:      Sample batch $B_t = \{(x_i, y_i)\}_{i=1}^N$
 4:      Update $\phi$: $\phi \leftarrow \phi - \eta_\phi \nabla_\phi \frac{1}{|B_t|} \sum_{(x_i, y_i) \in B_t} \text{MSE}(F_\phi(x_i) - y_i)$
 5: **end for**
 6: **Phase 2: train energy network**
 7: **for** $t = 1$ to $T$ **do**
 8:      Sample batch $B_t = \{(x_i, y_i)\}_{i=1}^N$
 9:      Generate $\tilde{y}_i = F_\phi(x_i)$ for $x_i \in B_t$
10:      Update $\theta$: $\theta \leftarrow \theta - \eta_\theta \nabla_\theta \frac{1}{|B_t|} \sum_{(x_i, y_i) \in B_t} [E_\theta(x_i, y_i) - E_\theta(x_i, \tilde{y}_i)]$
11: **end for**
12: **Phase 3: gradient-based inference**
13: Initialize $\tilde{y}_i^{(0)} = F_\phi(x_i)$ for $x_i \in B_t$
14: **for** $k = 1$ to $K$ **do**
15:      Refine $\tilde{y}_i$: $\tilde{y}_i^{(k)} \leftarrow \tilde{y}_i^{(k-1)} - \eta \nabla_{\tilde{y}} E_\theta(x_i, \tilde{y}_i^{(k-1)})$
16: **end for**

---

### 3.3 SETTING

**Datasets**  We conduct our experiments on Human3.6M 3D WholeBody dataset (H3WB) (Zhu et al., 2023) and Human3.6M dataset (H36M) (Ionescu et al., 2014). H36M is one of the most widely used datasets for 3D human pose estimation (Zheng et al., 2023; Liu et al., 2024). H3WB extends H36M by providing whole-body annotations using the COCO WholeBody layout, which includes 133 whole-body keypoint annotations, capturing detailed information about hands, face, and feet, making it suitable for tasks that require fine-grained pose estimation. We utilize the ground truth 2D joint locations provided in the dataset to align the 3D and 2D poses. For the H36M dataset, we zero-center the 3D poses around the pelvis joint, following standard protocols and prior work. For the H3WB dataset, we zero-center the 3D poses around the midpoint of the two hip joints.

**Implementation Details**  We employ the SimpleBaseline (Martinez et al., 2017), SemGCN (Zhao et al., 2019) and single frame version of VideoPose (Pavllo et al., 2019) as task-nets. For the H3WB dataset, we modify the input and output layers of these task-net to align with data. For the loss-net, we adjusted the SimpleBaseline by modifying the dimensions and depth of the hidden layers. We set the hidden size to 2048 with 2 residual block stages without batch normalization and dropout layers for H3WB and SimpleBaseline task-net for H36M. For the other task-net for H36M, we set the hidden size to 256 with 3 residual block stages with dropout layers. We use separate Adam optimizers (Kingma & Ba, 2015) without learning rate decay for the loss-net and the task-net. All models are trained with a batch size of 1024 for 50 epochs on H36M and a batch size of 64 for 200 epochs on H3WB. For hyperparameter tuning, we employed Bayesian optimization with the *wandb* sweep tool (Biewald, 2020), aiming to minimize MPJPE for the S9 and S11 in the H36M dataset and PA-MPJPE for the S8 in the H3WB dataset, following the convention of prior works. To avoid overfitting to a specific random seed, we reported the average results from experiments with different random seeds using the optimized hyperparameters.

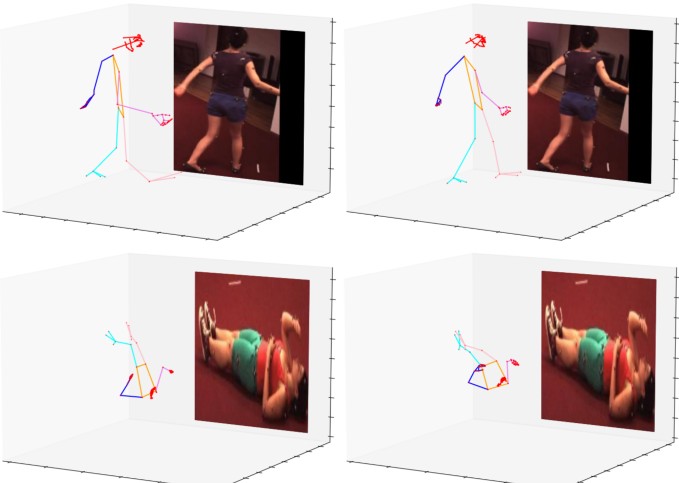

Figure 2: Comparison of outputs on the H3WB dataset: (left) Baseline method, (right) SEAL-Pose.

## 4 EVALUATION METRICS

We evaluate our models using standard metrics for 3D human pose estimation. For the H36M dataset, we report MPJPE and P-MPJPE (Procrustes-aligned MPJPE), following established protocols. On the H3WB dataset, we use the official benchmark's PA-MPJPE (Pelvis-aligned MPJPE), measuring errors for the whole body, body, hands, face, hands aligned on wrists, and face aligned on nose.

To further assess structural consistency in the predicted poses, we introduce three additional metrics:

### 4.1 LIMB SYMMETRY ERROR (LSE)

The Limb Symmetry Error evaluates left-right body symmetry by comparing the lengths of corresponding limbs on the left and right sides. It is defined as the normalized absolute difference in lengths between each pair of corresponding limbs.

Given a set of $n$ corresponding limb pairs, where the $i$-th left limb is defined by keypoints $\mathbf{l}_{i1}, \mathbf{l}_{i2}$ and the corresponding right limb by $\mathbf{r}_{i1}, \mathbf{r}_{i2}$, the LSE for limb pair $i$ is computed as:

$$\text{LSE}_i = \left| \frac{\|l_{i1} - l_{i2}\| - \|r_{i1} - r_{i2}\|}{\|l_{i1} - l_{i2}\| + \|r_{i1} - r_{i2}\|} \right|$$

where $\|\cdot\|$ denotes the Euclidean norm. A lower LSE indicates greater symmetry, which is desirable for anatomically plausible poses. We omitted the factor of $\frac{1}{2}$ for simplicity.

### 4.2 BODY SEGMENT LENGTH ERROR (BSLE) AND LIMB LENGTH ERROR (LLE)

The Body Segment Length Error measures deviations in the lengths of body segments—pairs of adjacent joints—by comparing predicted and ground-truth poses. For each segment $i$, with predicted keypoints $\mathbf{k}_{i1}, \mathbf{k}_{i2}$ and target keypoints $\mathbf{t}_{i1}, \mathbf{t}_{i2}$, BSLE is defined as:

$$\text{BSLE}_i = \left| \frac{\|\mathbf{k}_{i_2} - \mathbf{k}_{i_1}\| - \|\mathbf{t}_{i_2} - \mathbf{t}_{i_1}\|}{\|\mathbf{t}_{i_2} - \mathbf{t}_{i_1}\|} \right|$$

LLE is a specific case of BSLE focusing only on limb segments. This metric calculates the relative difference between predicted and target segment lengths, reflecting how well the model preserves anatomical proportions. Lower BSLE and LLE values indicate better preservation of segment lengths in the predicted poses.

Table 1: Performances on the H3WB dataset (MPJPE in mm). † from H3WB's official benchmark.

| Metric | Whole-body | Body | Face/Aligned | Hand/Aligned |
|---|---|---|---|---|
| SimpleBaseline† | 125.4 | 125.7 | 115.9 / 24.6 | 140.7 / 42.5 |
| Jointformer† (Lutz et al. (2022)) | 88.3 | 84.9 | 66.5 / 17.8 | 125.3 / 43.7 |
| 3D-LFM (Dabhi et al. (2023)) | 64.1 | 60.8 | 56.6 / 10.4 | 78.2 / 28.2 |
| SimpleBaseline | 65.5 | 62.8 | 49.6 / 14.6 | 92.7 / 35.1 |
| w/ Gradient-based Inference | 65.3 | 62.6 | 49.4 / 14.8 | 92.5 / 35.0 |
| w/ loss-net (margin) | **62.8** | **61.1** | **46.3 / 13.7** | **90.7** / 34.7 |
| w/ loss-net (NCE) | 63.4 | **61.1** | 46.5 / 14.5 | 92.1 / **34.2** |
| VideoPose | 60.1 | 56.4 | 46.3 / 11.9 | 84.3 / 29.6 |
| w/ Gradient-based Inference | 60.0 | 56.3 | 46.3 / 12.4 | 84.2 / 29.5 |
| w/ loss-net (margin) | **58.6** | 55.7 | **45.0** / 11.6 | **82.3** / 29.3 |
| w/ loss-net (NCE) | 58.8 | **54.8** | 45.5 / **11.5** | 82.7 / **28.9** |

Table 2: Performances on the H36M dataset (MPJPE in mm).

| task-net | SimpleBaseline | | SemGCN | | VideoPose | |
|---|---|---|---|---|---|---|
| metric | MPJPE | P-MPJPE | MPJPE | P-MPJPE | MPJPE | P-MPJPE |
| Baseline | 43.8 | 34.7 | 47.0 | 37.9 | 41.6 | **32.4** |
| w/ loss-net (margin) | **42.5** | 33.9 | **44.8** | **36.2** | **41.3** | **32.4** |
| w/ loss-net (NCE) | 42.7 | **33.8** | 44.9 | 36.5 | **41.3** | 32.5 |

## 5 EXPERIMENTAL RESULTS

### 5.1 POSE ESTIMATION ERROR EVALUATION

To evaluate the performance of SEAL-Pose, we assessed its impact on 3D whole-body pose estimation using the H3WB dataset, following the dataset's official PA-MPJPE benchmark for whole body, body, face, and hands. As detailed in Table 1, SEAL-Pose demonstrated substantial improvements across all body regions compared to baseline models. These reductions in error underline the framework's capacity to capture complex human body structures, including finer anatomical details like facial features and hand articulations. The improved performance validates SEAL-Pose's ability to model intricate interdependencies among body regions for more accurate and cohesive predictions. Notably, SEAL-Pose showed relatively better performance than the baseline on less common data distributions, such as target figures in reverse, lying down, or with partial occlusion, as illustrated in Figure 2.

To further assess the effectiveness of SEAL-Pose, we included a comparison with a gradient-based inference approach that directly utilizes the structured energy network. As presented in Table 1, the SEAL-Pose approach consistently outperformed the gradient-based inference approach. This indicates that integrating the structured energy network into the learning process, rather than using it solely for direct gradient-based optimization, is more effective.

Our approach was also evaluated on the H36M dataset, where it again outperformed the baseline, as shown in Table 2. However, the performance gains on H36M were more modest compared to H3WB. These results are due to the greater complexity of the H3WB dataset, which includes more dependencies among output variables and intricate body structures. The stronger performance on H3WB suggests that the SEAL framework is especially well-suited for tasks like 3D whole-body pose estimation, which involve more detailed and fine-grained predictions where capturing complex structures is crucial. Notably, SemGCN showed significant performance improvements on the H36M dataset, which will be further discussed in Section 5.3.

Table 3: Keypoint ratio exceeding structural consistency metrics error threshold on H3WB.

| Threshold | 0.1 | | | 0.2 | | |
|---|---|---|---|---|---|---|
| Metric | LSE | LLE | BSLE | LSE | LLE | BSLE |
| Ground Truth | 1.59% | - | - | 0.08% | - | - |
| SimpleBaseline | 8.17% | 23.28% | 20.59% | 1.25% | 6.31% | 4.40% |
| w/ loss-net | **7.10%** | **20.40%** | **19.76%** | **0.88%** | **4.46%** | **4.18%** |

Table 4: Keypoint ratio exceeding structural consistency metrics error threshold on H36M.

| Threshold | 0.1 | | | 0.2 | | |
|---|---|---|---|---|---|---|
| Metric | LSE | LLE | BSLE | LSE | LLE | BSLE |
| Ground Truth | 0.00% | - | - | 0.00% | - | - |
| SimpleBaseline | 3.27% | 10.74% | 20.30% | 0.51% | 1.96% | 2.19% |
| w/ loss-net | **2.54%** | **8.33%** | **15.92%** | **0.40%** | **1.37%** | **1.63%** |

## 5.2 POSE STRUCTURE EVALUATION

Additionally, we evaluated structural consistency by examining the proportion of keypoints with errors above the defined threshold across the entire validation set. Our method consistently showed lower error rates across all three structural metrics, as detailed in Table 3. Specifically, for the H3WB dataset, the percentage of keypoints exceeding a 0.2 error threshold in LSE was reduced from 1.25% to 0.88%, achieving a relative reduction of 29.6%. Similarly, LLE decreased from 6.31% to 4.46%, marking a relative reduction of 19.3%. As shown in Table 4, similar trends are observed for the H36M dataset. These findings highlight that SEAL's ability to capture structural parts well contributes to predicting more anatomically consistent and plausible 3D poses.

## 5.3 TRAINING DYNAMICS AND ENERGY NETWORK ANALYSIS

**Gradient-Based Inference for Energy Network Evaluation** We performed gradient-based inference with the structured energy network trained through SEAL-Pose while increasing the number of iterations to verify that it captures the pose structure. The results showed that the MPJPE, which measures simple errors relative to the ground truth, saturates after a few iterations, while the P-MPJPE, which evaluates errors after aligning coordinate transformations, decreases steadily over dozens of iterations, as illustrated in Figure 3. Since P-MPJPE is a more appropriate metric for evaluating the plausibility of the poses, this trend indicates that the gradient signals from the energy network gradually enhance the plausibility of the estimated poses.

**Task-Net Energy Levels Across Training Epochs** To evaluate the effectiveness of SEAL-Pose in exploiting the learning signals from the loss-net, we observed the energy levels of the task-net predictions at each training checkpoint, logged at every epoch. The energy metrics were compared between the baseline model and the SEAL-Pose model, as shown in Figure 3. During the training process, the task-net in SEAL-Pose demonstrated a more significant reduction in energy levels compared to the baseline, resulting in considerably lower energy values at the end of the training. The decrease in energy suggests that SEAL-Pose efficiently utilizes the structural consistency signals from the loss-net, resulting more coherent learning. The consistently reduced energy levels attained by SEAL-Pose support the hypothesis that the dynamic feedback from the loss-net enhances the training process, fostering a more structured and efficient optimization pathway.

**Training Stability** The incorporation of the loss-net in SEAL-Pose significantly improves the training stability of the inherently unstable SemGCN model, serving effectively as a regularizer in high-dimensional output spaces. In the H36M dataset, SemGCN exhibited notable training instability, highlighted by large fluctuations in MPJPE throughout the epochs. In contrast, SEAL-Pose

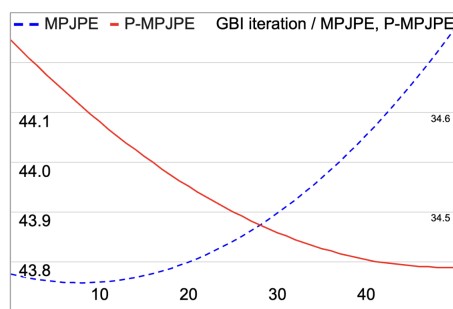 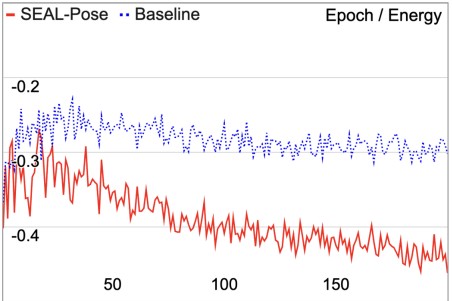

Figure 3: (left) MPJPE and P-MPJPE over GBI iteration. (right) Energy over training epoch.

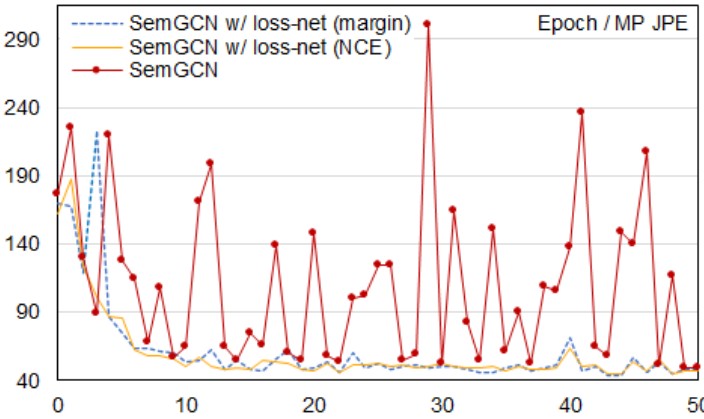

Figure 4: Training curves of SemGCN on the H36M dataset.

showed consistently stable training dynamics, regardless of the loss-net utilizing either a margin-based or NCE loss. The stability results from the joint optimization of the task-net and loss-net, encouraging a more smooth and consistent gradient flow. The enhanced gradient stability probably mitigates the challenges that models such as SemGCN encounter in capturing intricate interdependencies among body joints. As a result, SEAL-Pose not only stabilized the training process but also achieved substantial performance improvements for SemGCN on the H36M dataset, outperforming baseline.

## 6 CONCLUSION

In this work, we introduced SEAL-Pose, a novel adaptation of SEAL framework, specifically tailored for deterministic models in 3D human pose estimation. Our approach employs a structured energy network as a trainable loss function, effectively capturing joint dependencies and improving the coherence of predicted poses without relying on explicit structural priors. The application of SEAL-Pose demonstrated significant reductions in error on the H3WB and H36M datasets, producing more anatomically plausible poses compared to baseline methods. Furthermore, we introduced structural consistency metrics—Limb Symmetry Error (LSE) and Body Segment Length Error (BSLE)—to quantitatively evaluate pose plausibility, which highlighted the efficacy of our framework in capturing complex structures among body joints. Our findings highlight the potential of structured energy networks for enhancing tasks requiring complex output dependencies, such as 3D whole-body pose estimation, demonstrating that SEAL can be effectively extended to broader applications in the future.

## 7 LIMITATIONS

While SEAL-Pose demonstrates significant improvements in 3D human pose estimation, there are still areas for optimization and refinement. One key challenge lies in the broad hyperparameter search space, which includes weights for the energy loss term, learning rates for the task-net and loss-net, and other architecture-specific parameters. This extensive search space can make the optimization process computationally intensive and less straightforward. Identifying more efficient strategies for hyperparameter tuning could enhance the practicality and performance of the approach. Additionally, the effectiveness of SEAL-Pose could be further improved by developing more sophisticated loss-net architectures capable of providing stronger and more targeted learning signals for the task-net. Furthermore, while our experiments highlight that SEAL-pose was effective on the H3WB and H36M datasets, testing across various task-net architectures and diverse datasets would strengthen our claims.

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
