# OpenReview forum: "SEAL-Pose: Enhancing Pose Estimation through Trainable Loss Function"
_ICLR.cc/2025/Conference — Submitted to ICLR 2025_

### Official Review · Reviewer_1iNt · 2024-10-28

**Soundness:** 2
**Presentation:** 2
**Contribution:** 2
**Rating:** 3
**Confidence:** 5

**Summary:**

The paper introduces a Structured Energy As Loss (SEAL) framework aimed at improving 3D human pose estimation. However, the paper is incomplete (less than eight pages) and lacks a description of the method’s motivation, a comparison with prior approaches, and an evaluation against state-of-the-art methods.

**Strengths:**

The proposed method is simple but effective.

**Weaknesses:**

1. The diagram in Figure 1 is overly simplistic and does not effectively convey the motivation behind the proposed method.
2. The paper is incomplete (less than eight pages), lacks a thorough description of the method’s motivation, a discussion on advantages over prior methods, and comparisons with state-of-the-art approaches.

**Questions:**

Please see the Weaknesses.

---

> ### Author Response · Authors · 2024-11-26
>
> We are thankful for your feedback. We have taken your concerns into account and have revised our manuscript accordingly to strengthen the presentation of our work.
> - The intent of Figure 1 was to illustrate the architecture of the loss-net. However, we acknowledge that it may not fully capture the motivation behind the proposed method. In the revised manuscript, we have added additional details and an expanded description to clarify the interplay between the task-net and loss-net and to better illustrate the core ideas of SEAL-Pose.
> - While our selected baselines (e.g., VideoPose) are still competitive in monocular pose estimation, we acknowledge that comparisons with more recent state-of-the-art approaches would strengthen the evaluation and we have added the point to the limitation section in the revised manuscript.
> - Regarding the paper length, we followed the new ICLR guideline encouraging submissions of six or more pages. We have also made detailed descriptions and overall improvements to the discussions on your concerns, include motivation, advantages and comparisons, in the revised manuscript.

---

> > ### Comment · Reviewer_1iNt · 2024-11-26
> >
> > I have read the rebuttal and other comments. The submission is a rough draft, not a complete version. It should be revised according to the reviewers' comments and then resubmitted to the next conference.

---

### Official Review · Reviewer_US56 · 2024-11-03

**Soundness:** 2
**Presentation:** 1
**Contribution:** 3
**Rating:** 5
**Confidence:** 4

**Summary:**

This paper presents SEAL-pose to handle 2D-to-3D lifting problem in human pose estimation. The core is training a learnable loss function to encode the structure of output body limbs. To evaluate the performance, the authors propose two new metrics: Limb symmetry error (LSE) and Body segment length error (BSLE). Experiments are performed on H36M and H3WB.

**Strengths:**

The incorporation of SEAL framework for 3D human pose lifting is a novel idea. The experiments could support the effectiveness of SEAL-psoe to some degree.

**Weaknesses:**

Description of methods should be improved. Experiments are not enough. See Questions below.

**Questions:**

1. The technical part is not clearly stated, thus making the readers hard to follow. For example,
in eqn(1),(3), how is B_t defined? How is \eta_{\phi} defined? Is “x” the input image or 2D pose for lifting? Is L the joint number?

2. There are many previous methods that estimate a probabilistic distribution of output 3D skeleton from 2D input (e.g. Diffusion-based 3d human pose estimation with multi-hypothesis aggregation, ICCV 2023). It may be better for SEAL pose to combine with such framework instead of a deterministic model.

3. Why not use MPI-INF-3DHP for evaluation? The experiments have been limited to the camera settings of H36M, which may not convince the readers for the generalization ability of SEAL-pose.

4. The authors propose LSE and BSLE. However, they only report these two metrics on H3WB in Table.3. Why not report on H36M for body only poses?

5. Is there any visualized comparison to demonstrate the 3d lifting quality of SEAL-pose? Without visualized results, I could not fully understand the improvement brought by lower LSE and BSLE. As H3WB contains 133 keypoints with distinct scale, which part is  improved the most (face, or hand, or body)?

---

> ### Author Response · Authors · 2024-11-26
>
> We are truly grateful for your thorough review and valuable suggestions. We have carefully considered and addressed the points you raised, and your feedback helped a lot in the revision of our paper. We also address your concerns below.
> 1. We revised the manuscript to clarify the definitions of the variables in the equations to improve  readability and ensure technical accuracy in the equations.
> 2. We agree that integrating SEAL with probabilistic frameworks, such as those estimating distributions, could be a valuable extension. Our focus, however, was to demonstrate the utility of SEAL in deterministic models, with 3D HPE serving as a practical testbed for this purpose. We will emphasize this intent in the revised manuscript.
> 3. While we prioritized H3WB due to its complexity and inclusion of 133 keypoints, we recognize the value of using additional datasets such as MPI-INF-3DHP to demonstrate broader generalizability. Therefore, we have added this to the limitations section and plan to explore this direction in future work.
> 4. Thank you for pointing out the omission of LSE and BSLE results for H36M. We have now included these metrics for H36M in Table 4 of the revised manuscript.
> 5. Due to the limitations of 3D HPE visualization, it is difficult to confirm the differences (by visualization) in these metrics dramatically. However, we have improved the description of LSE and BSLE metrics in the revised manuscript.

---

> > ### Comment · Reviewer_US56 · 2024-11-26
> > **I would like to maintain the original rating**
> >
> > The revised manuscript has improved a lot. However, using only H36M could not convince broader readers. Therefore I maintain the rating, and wish the authors test on more datasets to prove the efficacy of SEAL-pose.
> >
> > Also, I have additional questions. I notice that the quantitative numbers of Table.1 and Table.2 change a lot, yet the training curve in Figure. 4 remains the same. What happened for such improvements? And what  can we learn from the comparison between loss-net(margin) and loss-net(NCE)? Their results are quite similar and no further explaination was given.

---

> > > ### Author Response · Authors · 2024-11-28
> > >
> > > Thank you for your careful and thorough review.
> > > - The results in Tables 1 and 2 have changed due to additional hyperparameter searches. Specifically, the SEAL-Pose results for the H3WB dataset and the results of applying SEAL-Pose to the SimpleBaseline model for the H36M dataset have changed. However, the SemGCN experimental results depicted in Figure 4 remain unchanged. Instead, we identified that the right chart in Figure 3 needed to be changed, so we have updated it.
> > > - In addition, margin-based loss and NCE loss are the loss functions for training the loss-net we used, following the prior work. The importance of which loss is used was not prominent in our experiments, however, one of our next steps will be to analyze which of them works better, and particularly in what situations.

---

### Official Review · Reviewer_wPPr · 2024-11-03

**Soundness:** 2
**Presentation:** 2
**Contribution:** 2
**Rating:** 3
**Confidence:** 4

**Summary:**

This work addresses 2D-to-3D human pose lifting, by applying the "Structured Energy as Loss" (SEAL) framework. That is, besides training a regressor for the task (here called task-net), a second network (loss-net) is also trained that scores prediction-input pairs for their plausibility. At inference time, the output can be taken as is, or gradient-based inference (GBI) can be applied to modify the prediction towards driving down the loss-net output (to make the prediction seem more plausible to the loss-net). This GBI can be applied by tuning either the prediction itself or the model weights.
The motivation is to better take into account dependencies in the output variables, such as bone lengths in the case of human pose estimation.
The model is evaluated on Human3.6M and its whole-body extension and is shown to improve the scores.

**Strengths:**

* The problem of human pose lifting is of interest to the research community.
* Applying the SEAL framework to this task is novel.
* The method, when applied to SimpleBaseline and VideoPose3D, improves results on Human3.6M, both in terms of the standard metrics and in terms of bone symmetry and bone length error.
* The ablation model (Section 3.2) is useful in demonstrating the value in training the loss-net together with the task-net.

**Weaknesses:**

* The most serious problem I see is using a hyperparameter sweep tool (from WandB) to tune hyperparameters directly for the Human3.6M/H3WB test set. If this was indeed done so, it invalidates the seen improvements, as the gap is not so large and tuning hyperparameters for a particular test set can aways achieve significantly stronger results.
* While the method is motivated from the prior SEAL method, I am not convinced that the terminology of structured energy is helpful for understanding rather than obscuring what actually happens in the approach. In essence, the final model is a fairly standard conditional GAN. The terminology of "structured" learning/energy/outputs comes from the pre-deep-learning era when multi-output model were generally less standard and more difficult to train. Today, talking about a learned loss function or indeed simply a discriminator may be much better understood by the community, since models that make "structured" predictions are commonplace today. GANs have been used in a similar fashion for 2D-to-3D lifting (e.g. [3]), diminishing the novelty of the proposed paper.

* The work only uses Human3.6M (and its wholebody extension). While Human3.6M has been very valuable for research over the last decade, today we now have many more datasets and it is now possible to give stronger evidence for a method than improvements on two specific subjects. The particular bone-length structures of subject 9 and subject 11 of H36M may not generalize. Other possible training datasets would include the following (of course I do not expect using all of these, it is just to give some ideas): MPI-INF-3DHP, CMU-Panoptic, AMASS, HuMMan, AIST-dance++, AGORA, BEDLAM, GeneBody, DNA-Rendering, RICH etc. And other evaluation datasets include 3DPW and EMDB. Again, I do not expect using all, but using at least something further beyond the Human3.6M data would make the evidence much stronger.
* The baseline methods (SimpleBaseline, VideoPose) are fairly old by the standards of this field (2017 and 2019). It would be important to try newer lifting methods as well. The results are also not compared to the current SOTA methods or those from the last five years.
* The writing is quite verbose, for example an full page (page 5) is spent on describing limb length losses, which could be expressed in a briefer way. Such bone-based losses and metrics have been used in many prior works, for example [2,3]
* It is not clear why the SemGCN model has spikes in the training curve (Fig. 3). These spikes can appear for many practical reasons in a particular implementation and do not generally mean a fundamental problem.

**Questions:**

* Is GBI used in combination with SEAL? Fig. 2 (left) seems to indicate so but the text does not make this clear.
* As far as I understand, the method is only tested with ground-truth 2D keypoints (L200). This is not a realistic setup, and 2D keypoints from 2D pose estimators would yield a more realistic evaluation. Is this correct?
* Are the hyperparameters tuned for the test set?

---

> ### Author Response · Authors · 2024-11-26
>
> We greatly appreciate the time and effort you have put into reviewing our paper. Your feedback has been invaluable, and we have made sure to address your questions and concerns to enhance the quality of our work. We respond to your concerns below.
> - GBI is applied as described in Section 3.2. to train the energy network using baseline predictions as negatives. This evaluation contrasts the effectiveness of structured energy networks as direct predictors (GBI) versus learning signal providers (regularizers), as in our proposed method. We have made this section's description more understandable.
> - Following the convention of prior works using Human3.6M, we optimized and validated models on S9 and S11. We acknowledge the concern and have clarified this in revised manuscript to emphasize adherence to standard practices. We have also condensed verbose section that you mentioned.
> - One of our main point is SEAL-Pose more benefit on the complex dataset such as H3WB, and the direct comparison to Human3.6M is most proper to validate the claim. Therefore, we prioritized Human3.6M and its whole-body extension (H3WB). In addition, regarding monocular pose estimation, our baselines (e.g., VideoPose) are still competitive methods. However, we agree that evaluating on diverse datasets would strengthen the generalizability of our method, therefore we have added this point to the limitations section at the end of the manuscript.
> - While our framework shares conceptual similarities with conditional GANs, there are critical distinctions. Our task-net and loss-net do not compete to maximize and minimize a shared objective. Instead, the loss-net learns to capture structured output dependencies by leveraging a NCE loss or margin-based loss, which allows task-specific flexibility. For instance, unlike conditional GANs, our margin-based loss enables explicit control over the loss-net's learning focus by setting task-relevant margins (e.g., MPJPE in our experiments). This ensures that the loss-net captures meaningful structural dependencies related to the margin function.
> - There is an expression in your comment that seems to cite a paper, but there is no list of related papers, so if there is something missing, please let us know so that we can refer to it.

---

> > ### Comment · Reviewer_wPPr · 2024-11-26
> >
> > Optimizing hyperparameters on the final evaluation subjects S9 and S11 will distort the results. At this point only adding further benchmarks (which is common practice), such as MPI-INF-3DHP, would help in assessing the strength of the method.
> >
> > I also agree with the other reviewers that the writing quality is not mature enough at this stage. I believe that incorporating the outcome of the discussion will help make the paper stronger, but this would not fit the timeline of this conference and would need another review.
> >
> > (I apologize, the citations were left off
> > [2] Enhancing 3D Human Pose Estimation with Bone Length Adjustment https://arxiv.org/pdf/2410.20731
> > [3] Xin Cao and Xu Zhao. Anatomy and Geometry Constrained One-Stage Framework for 3D Human Pose Estimation. ACCV 2020)

---

### Official Review · Reviewer_ovua · 2024-11-05

**Soundness:** 2
**Presentation:** 2
**Contribution:** 2
**Rating:** 3
**Confidence:** 4

**Summary:**

This paper presents SEAL-Pose, a Structured Energy As Loss framework to improve the 3D pose estimation from 2D keypoints. The SEAL loss is previously applied in probabilistic models whereas the paper modifies it to deterministic models. This loss improves the dependencies among the keypoints which produces more plausible poses empirically.

**Strengths:**

1. This paper addresses an interesting problem of improving the plausibility of output pose. The SEAL loss is new and effective.
2. This paper is easy to understand.

**Weaknesses:**

1. The writing of this paper can be improved, e.g., the names of task-net and loss-net are confusing since the task here is only pose estimation and loss-net only refers to energy loss. Also, it would be clearer if authors can give an overview of what each net functions in the beginning of Sec.3.1.
2. The motivation is not clearly explained and verified. It’s not clear why SEAL-Pose can improve the plausibility of estimated poses and why it can perform better in the proposed metrics, i.e., LSE, BSLE, and LLE.
3. The method should be evaluated on more advanced frameworks. It’s necessary for this method to compete against state-of-the-art methods and on more challenging benchmarks, for example, 3DPW, MPI-INF-3DHP. In Table 2, the performance is very similar to the baseline method VideoPose.
4. Minor issues:
    1. Inconsistency between P-MPJPE and PA-MPJPE in H36M and H3WB
    2. The quality of figures can be improved

**Questions:**

The contributions do not reach the bar of ICLR therefore I recommend rejection. The further improvement can be better presentation of paper and more experimental validation of the proposed loss.

---

> ### Author Response · Authors · 2024-11-26
>
> We are thankful for your detailed feedback. We have carefully addressed your concerns, and your review has been extremely helpful in improving the clarity and depth of our manuscript.
>
> 1. We followed the SEAL paper's terminology (task-net, loss-net) to maintain consistency with prior literature. We revised Section 3.1 to provide clearer descriptions of their respective roles and functions.
> 2. Our motivation is rooted in the structured energy network's (loss-net) ability to capture dependencies that are not effectively captured during task-net training by optimizing only to minimize MSE. We hypothesized that loss-net can provide a useful learning signal for the task-net, enhancing its output structures. Our proposed metrics (LSE, BSLE, and LLE) quantify the structural plausibility of the predicted poses, confirming the effectiveness of SEAL-Pose. Based on your comments, we have improved the paper overall.
> 3. We conducted experiments on the H3WB dataset, which involves more joints and a complex structure than the H36M or similar datasets, comparing against recent state-of-the-art (SOTA) models. While our selected baselines (e.g., VideoPose) are still competitive in monocular pose estimation, we agree that evaluating on diverse datasets and task-net architectures would strengthen our claims, therefore we have added this point to the limitations section at the end of the manuscript.
> 4. The difference arises because P-MPJPE on H36M follows the commonly used "Protocol #2" (Procrustes-aligned MPJPE), while PA-MPJPE on H3WB, proposed by its authors, is pelvis-aligned MPJPE. These definitions differ by benchmark conventions.

---

> > ### Comment · Reviewer_ovua · 2024-11-26
> > **I prefer to maintain my original score.**
> >
> > Thank the authors for their explanations! I appreciate that the authors revise Sec 3.1 to give better explanations of task-net and loss-net. However, it's still not clear why the loss-net is able to improve the plausibility of the human pose. Also, I agree with Reviewer 1iNt that it's not an established manuscript. I suggest the authors can revise in the future and submit to upcoming venues.

---

### Author Response · Authors · 2024-11-26

We would like to sincerely thank the reviewers for their thoughtful comments and constructive feedback on our manuscript. Their insights have greatly contributed to improving the quality of our work. We responded to reviewers' questions and concerns in comments, and we have updated our paper overall to reflect their feedback. Therefore, we would really appreciate it if the reviewers could check our responses and revised manuscript.

---

### Author Response · Authors · 2024-12-01

We are very grateful for the thoughtful reviews. Because reviewers all gave us similar feedback, we did additional experiments on MPI-INF-3DHP, a more challenging dataset, and the results are shown in Table 1. Performance improvements through SEAL-Pose are more pronounced on this dataset.

We hope this could support our claims and verify the effectiveness of our method. We are glad to have had the opportunity to enhance our work through constructive discussions and offer our gratitude to the reviewers.

**Table 1. Performances on the MPI-INF-3DHP dataset (MPJPE in mm).**
| task-net             | SimpleBaseline |          |  SemGCN  |          | VideoPose |          |
|----------------------|:--------------:|:--------:|:--------:|:--------:|:---------:|:--------:|
| metric               |      MPJPE     |  P-MPJPE |   MPJPE  |  P-MPJPE |   MPJPE   |  P-MPJPE |
| baseline             |      80.9      |   63.9   |   74.5   |   57.9   |    66.4   |   50.8   |
| w/ loss-net (margin) |    **71.8**    | **54.9** | **71.8** |   55.8   |    64.1   | **48.2** |
| w/ loss-net (NCE)    |      72.3      |   56.0   |   72.5   | **54.7** |  **64.0** |   48.3   |

---

### Meta-Review · Area_Chair_crCE · 2024-12-21

**Metareview:**

This paper addresses 2D-to-3D human pose lifting, by applying the "Structured Energy as Loss" (SEAL) framework. The SEAL loss is previously applied in probabilistic models whereas the paper modifies it to deterministic models. Besides training a regressor for the task (called task-net), a second network (loss-net) is also trained that scores prediction-input pairs for their plausibility. At inference time, the output can be taken as is, or gradient-based inference (GBI) can be applied to modify the prediction towards driving down the loss-net output. Experimental results on the Human3.6M and Human3.6M WholeBody datasets show that effectiveness of the proposed method.  While applying the SEAL framework to 3D human pose estimation is appreciated, the reviews are unanimously negative.  The reviewers raised concerns regarding limited validation and poor writing.  Evaluation on only Human3.6 (and its wholebody extension) is insufficient.  Even if Human3.6M has been recognized as valuable dataset for over the last decade, many more datasets are now available. Moreover, comparisons with recent SOTAs are also missing.  Regarding writing, motivation is not clearly explained.  There are verbose descriptions as well.  The authors addressed, in the rebuttal, the raised concerns.  As a result, the writing is improved a lot.  But evaluation on other datasets and comparison with SOTAs are not provided but just discussed as the limitation, leaving a question on the generalization ability of the proposed method.  The reviewers have felt that evaluation using only Human3.6M cannot convince broader readers and evaluation on more datasets is required to prove the effectiveness of the proposed method.  This paper should be rejected, accordingly.

**Additional Comments On Reviewer Discussion:**

See above.

---

### Decision · Program_Chairs · 2025-01-22

Reject